# Selected Lipidome Components and Their Association with Perinatal Depression

**DOI:** 10.3390/nu17223590

**Published:** 2025-11-17

**Authors:** Dominika Ładno, Beata Nowak, Aleksandra Palka, Dominik Strzelecki, Oliwia Gawlik-Kotelnicka

**Affiliations:** 1Faculty of Medicine, Medical University of Lodz, 90-647 Lodz, Poland; dominika.ladno@student.umed.lodz.pl (D.Ł.); beata.nowak1@student.umed.lodz.pl (B.N.); aleksandra.palka@student.umed.lodz.pl (A.P.); 2Department of Affective and Psychotic Disorders, Medical University of Lodz, 90-647 Lodz, Poland; oliwia.gawlik@umed.lodz.pl

**Keywords:** perinatal depression, lipidome, lipids

## Abstract

**Background/Objectives**: Perinatal depression affects approximately 21% of pregnant women and 15% postpartum, significantly impacting both maternal and child health. Lipid metabolism alterations, particularly involving fatty acids and lecithin, have been associated with mood disorders during the perinatal period. Omega-3 PUFAs (polyunsaturated fatty acids) play a key role in mood regulation and neuroinflammatory processes, while lecithin significantly influences neurotransmitter synthesis. **Methods**: A narrative review was conducted using PubMed, Scopus and Google Scholar for relevant articles which were qualitatively analyzed. Most of the literature included was published between 2020 and 2025 with selected earlier studies used, primarily, to outline the theoretical background. **Results**: This narrative review highlights substantial evidence linking components of lipidome, particularly omega-3 fatty acids and lecithin, and the occurrence of perinatal depression. Omega-3 deficiency increases antenatal depression risk by up to 6-fold. Inflammation, manifested by elevated levels of inflammatory markers (interleukin-6, tumor necrosis factor, C-reactive protein), and kynurenine pathway activation appear as central mechanisms, both of which can be modulated by PUFAs. Supplementation shows variable outcomes, with greatest efficiency for eicosapentaeonic acid (EPA)-predominant formulations (EPA/DHA ≥ 1.5). Choline is essential for fetal neurodevelopment, though evidence on lecithin and choline is inconclusive. Presumably, excessive intake and trimethylamine *N*-oxide (TMAO) production may contribute to depressive symptoms. **Conclusions**: Omega-3 PUFAs deficiency may increase the risk of perinatal depression, while supplementation appears beneficial for prevention. The findings regarding other lipid-derived compounds, specifically choline and lecithin, are inconclusive. Despite promising findings, further research is necessary to confirm the effectiveness of dietary interventions.

## 1. Introduction

Perinatal depression, which includes antepartum and postpartum depression, is defined as a non-psychotic depressive episode of varying intensity, from mild to severe, which emerges during gestation or within the first year after labor [1]. It presents a significant challenge in the clinical practice of obstetricians and psychiatrists. Studies suggest that about one in five women (20.7%) experience depression during pregnancy, while more severe symptoms affect around one in six women (15.0%) [2]. Rates in low- and middle-income countries tend to be higher at 24.7% [3]. The prevalence of postpartum depression also varies according to country, with the highest rates in developing countries, with a mean prevalence rate of 14.0% [4]. 

However, detection and treatment rates remain suboptimal. Research published in 2016 shows that only 49.9% of women with depression during pregnancy and 30.8% of those with postpartum depression are identified, and only 13.6% of women suffering from antenatal depression and 15.8% of women with postpartum depression receive appropriate treatment [5]. Notably, a large proportion of antidepressants are regarded as safe for use during lactation and breastfeeding, with sertraline being recommended as first-line medication. Obstacles to receiving proper treatment encompass inadequate availability of specialists and stigma, among others [6]. 

It is important to highlight that antenatal depression and anxiety are considered as one of the strongest risk factors for postpartum depression [7] and more than 50% of women who suffer from antepartum depression will eventually develop postpartum depression [8]. Not only does perinatal depression affect women, causing their profound suffering, but it also impairs the child’s socio-emotional, cognitive, linguistic, and motor development, with potential long-term effects into adolescence [9]. A systematic review of 20 population-based contemporary cohort studies showed that untreated antenatal depression is significantly linked to preterm birth, low birth weight, stillbirth, and maternal morbidity, as well as the occurrence of autism spectrum disorder [10]. Antepartum depression is correlated with obstetric complications, such as reduced 1 and 5 min APGAR scores, the prevalence of pharmacological induction of labor and child admission to neonatal intensive care unit, as well as increased likelihood of failed breastfeeding [11]. 

In terms of biological factors, the increased risk of these affective disorders during pregnancy and the postpartum period is closely associated with changes in fatty acid levels, steroid hormone levels and the activity of the kynurenine pathway [12]. In many international centers, including China [13,14], Brazil [15], and Japan [16], an association between metabolomic alterations in pregnant women and mental disorders both before and after labor was found. Most research focuses on amino acids, while lipid components also serve essential roles in the body, including building cell membranes and taking part in signal transduction [17]. They serve a key function in the nervous system, including aiding in the development of the fetal brain [18,19]. Their role has already been acknowledged in major depressive disorder in both rodent models [20] and clinical studies [21].

This review aims to discuss and summarize the impact of the aforementioned metabolomic components on the occurrence of perinatal depression, focusing on lipid components. The findings may provide a better understanding of the pathophysiological mechanisms of perinatal mood disorders, facilitating early detection and consideration of introducing supplementation and other dietary interventions (Figure 1).

## 2. Materials and Methods

This manuscript was developed as a narrative literature review aimed at synthesizing current knowledge on the relationship between lipidome components and perinatal depression. The review focused on epidemiological, clinical, biochemical, and mechanistic studies addressing lipid metabolism, fatty acids, phospholipids, ceramides, choline-containing compounds, and lipid-related biomarkers in the context of perinatal and postpartum depression.

### 2.1. Literature Search Strategy

A structured literature search was performed using PubMed, Scopus, and Web of Science databases. The search covered publications from 2007 to 2025 in order to include both foundational studies and recent advances in the field. The following keywords and their combinations were used: “perinatal depression,” “postpartum depression,” “lipid metabolism,” “lipidomics,” “fatty acids,” “omega-3,” “omega-6,” “ceramides,” “choline,” “lecithin,” “lipid biomarkers,” “inflammation,” and “tryptophan-kynurenine pathway.”

### 2.2. Inclusion and Exclusion Criteria

Eligible studies included peer-reviewed original articles, systematic reviews, meta-analyses, cohort and case–control studies, clinical trials, and narrative reviews that examined the association between lipid metabolism (or specific lipid components) and perinatal depression. Studies in humans were prioritized, although selected mechanistic and translational studies in animal models were also included to provide insight into biological pathways. Articles were restricted to those published in English. Exclusion criteria were conference abstracts, editorials, non-peer-reviewed sources, and studies unrelated to perinatal depression or lipid biology.

### 2.3. Study Selection and Data Extraction

Titles and abstracts were screened for relevance, followed by full-text review. Data extracted from eligible publications included the following: study type, population characteristics, biochemical or clinical markers assessed, methods for lipid or metabolomic analysis, prevalence or severity of perinatal depression, and reported associations between lipidome components and psychiatric outcomes. Special attention was given to omega-3 and omega-6 fatty acids, choline and lecithin metabolism, ceramide signaling, tryptophan–kynurenine pathway interactions, inflammatory biomarkers, and neurotrophic factors (e.g., BDNF, VEGF).

### 2.4. Data Synthesis

Extracted data were synthesized narratively and thematically. Findings were categorized into (1) epidemiological prevalence studies, (2) clinical trials and meta-analyses on lipid-based interventions, (3) biomarker and metabolomics studies, and (4) mechanistic studies exploring lipid-related pathways in depression. Cross-study comparisons were made to highlight consistencies, discrepancies, and emerging trends. In total, 101 scientific papers were analyzed. Of these, 67 publications were used to provide background scientific information, definitions, and theoretical context. Forty-one publications were included in the discussion and narrative synthesis of this article. Among the studies used for the discussion, 2 were systematic reviews, 3 were meta-analyses, 13 were narrative reviews, 19 were original research articles (including 5 cohort studies, 1 case–control study and 4 observational studies), 3 were clinical trials, and 1 was a book chapter (Figure 2).

## 3. Results and Discussion

### 3.1. Changes in Lipid Metabolism During Pregnancy

Given the complex etiology of perinatal depression, understanding metabolic changes during pregnancy offers insight into potential risk factors and mechanisms, particularly those related to lipid metabolism.

During pregnancy, maternal lipid metabolism demonstrates dynamic changes. The first and second trimesters are collectively referred to as the anabolic phase. Elevated levels of estrogen, progesterone, and insulin enhance lipid storage and suppress lipolysis, while adipose tissue deposition is supported by GH and prolactin [22]. Among women with body mass index (BMI) within the normal range (18.5–24.9), expected weight gain is 25 to 35 lbs (around 11–16 kg) [23]. During this phase, biosynthesis of free fatty acids and lipoprotein lipase expression increase, leading to gradual elevation of triacylglycerols, cholesterol, and phospholipids [24].

At the 30th week of pregnancy, a metabolic shift occurs as the catabolic phase begins, during which glucose and amino acids are preserved for the developing fetus, and lipids become the primary maternal source of energy [25]. This state is characterized by insulin resistance and decreased lipoprotein lipase activity. Lipolysis is enhanced by the increase in human placental lactogen [26]. In physiological gestational hyperlipidemia, levels of TAGs elevate by 50 to 100%, LDL concentration rises by 30 to 50%, and HDL levels increase by 20 to 40% [22]. Plasma levels of phospholipids also increase by around 65% compared to the initial phase of pregnancy [26].

These metabolic changes during pregnancy support fetal development, but they may also influence maternal mental health, suggesting that disruptions in lipid homeostasis could be linked to mood changes during the perinatal period.

### 3.2. Fatty Acids

Among the various components of lipid metabolism, fatty acids play a particularly significant role due to their direct involvement in brain function and mood regulation (Table 1).

Fatty acids, specifically polyunsaturated fatty acids (PUFAs), including omega-3 and omega-6, play a vital role in the building and functioning of the nervous system. As they are not biosynthesized in mammalian cells, α-linolenic acid (ALA, 18:3*n*-3) and linoleic acid (18:2*n*-6)—which are precursors of *n*-3 and *n*-6 PUFAs—must be supplied by dietary intake, along with eicosapentaenoic acid (EPA), docosahexaenoic acid (DHA) and arachidonic acid (AA), which can also be obtained directly from the diet [27].

The most commonly present PUFAs in the grey matter of the brain are *n*-3 PUFA docosahexaenoic acid (DHA) and *n*-6 PUFA arachidonic acid (AA, 20:4*n*-6). ALA, eicosapentaenoic acid (EPA, 20:5*n*-3), and docosapentaenoic acid (DPA, 22:5*n*-3) can cross the blood–brain barrier, but they are quickly oxidized, therefore constituting less than 1% of the brain’s fatty acids; however, unesterified DHA can accumulate in the grey matter. While *n*-3 PUFA deficiency is proven to decrease DHA levels and increase the AA/DHA ratio in the brain, lower dietary *n*-6 PUFA intake causes both the AA level and the AA/DHA ratio to decline. Studies conducted on rodent brains suggest that a decreased concentration of *n*-3 PUFAs, which is associated with a higher AA/DHA ratio, is associated with changes in synaptic function and resilience, which may contribute to neurogenesis impairment and behavioral abnormalities [27]. An elevated omega-6/omega-3 ratio has been associated with increased systemic inflammation and greater risk of depressive symptoms, with elevated IL-6 and TNF-α levels [28,29]. In general, PUFAs act in aiding neuronal transmission, maintaining cell membrane stability and elasticity, and ensuring effective signal transduction, which are all essential for cognitive and emotional functioning [30].

There is substantial evidence suggesting that lower levels of omega-3 fatty acids contribute to the development of depressive disorders, and even more antenatal and postpartum depression. Lower levels of omega-3 PUFAs have been shown to significantly contribute to the occurrence of perinatal depression, with affected women being up to six times more likely to experience antenatal depression compared to those with higher omega-3 levels [31]. In contrast, while omega-3 supplementation has demonstrated some efficacy in treating major depressive disorder (MDD) in the general population, its effects appear to be more pronounced in perinatal women, particularly when the EPA/DHA ratio is ≥1.5 [32,33]. These findings indicate that omega-3 deficiency may have a greater impact on perinatal depression than on depressive disorders in the general population.

The demand for DHA increases during pregnancy, as it is a critical constituent for fetal brain development [34]. Fetal DHA intake sums to 30–45 mg daily in the last trimester of gestation, while AA accumulation is mostly postnatal [35]. This increased demand can deplete maternal omega-3 stores, potentially increasing vulnerability to mood disorders, including antenatal and postpartum depression. A low concentration of serum omega-3 can be associated with a higher prevalence of antenatal depression [36]. A 2019 study by Hoge et al. suggests that a lower level of serum DHA can negatively impact maternal health with a higher prevalence of postpartum depression [37]. These outcomes positively correlate with the finding that a greater dietary intake of fish, which contain high levels of *n*-3 PUFAs, can act as protection from postpartum depression [38]. In contrast, a 2020 study by Mocking et al. proves that omega-3 supplementation has not been efficient in preventing perinatal depression and has been moderately efficient in treating perinatal depression, especially in women diagnosed with postpartum depression, which highlights the need for further, better-designed studies [39].

#### 3.2.1. Inflammation, Perinatal Depression and Their Connection to Fatty Acids

While fatty acids influence mood regulation, their connection to inflammatory processes further complicates the pathophysiology of perinatal depression, highlighting the need to consider both aspects at the same time.

Emerging evidence indicates that inflammation plays a critical role in the development of perinatal depression. Studies have identified elevated levels of inflammatory markers, such as interleukin-6 (IL-6), interleukin-8 (IL-8), C-reactive protein (CRP), and tumor necrosis factor-alpha (TNF-α), in women experiencing depressive symptoms during pregnancy and postpartum [40]. These markers, alongside kynurenine pathway metabolites and interleukin-1β (IL-1β), are linked to increased depression severity and may even predict future depressive episodes [41]. The activation of the kynurenine pathway, decreased T-cell activity, and the initiation of the NLRP3 inflammasome have been highlighted as key contributors to postpartum depression [42].

Recent research has provided further insight into how omega-3 fatty acids influence the kynurenine pathway, a metabolic route of tryptophan, which is known to be dysregulated in depression [43]. The kynurenine pathway is the primary pathway of tryptophan metabolism, in which indoleamine 2,3-dioxygenase (IDO-1, IDO-2) and tryptophan 2,3-dioxygenase (TDO) convert tryptophan into kynurenine and further metabolites. IDO is largely presented in the central nervous system’s immune cells and is known to be activated by inflammation and immune response [44,45], which have been linked to both depression in the general population [46] and, as stated before, perinatal depression. TDO activity, on the other hand, may be heightened by increased glucocorticoid levels, more specifically cortisol [47]. Considering cortisol levels are known to be raised in depressive disorders [48], this highlights the importance of the kynurenine pathway in the pathophysiology of depression.

In contrast, some studies have shown that TDO activity may be increased only by temporarily heightened cortisol levels and lowered by a chronic increase in cortisol. However, they suggest that a chronically high cortisol level can impair the activity of tryptophan hydroxylase (TPH), thus negatively affecting serotonin production [49].

Activation of the kynurenine route leads to the production of neuroactive metabolites, some of which can be neurotoxic. Some examples are quinolinic acid (QA), which can cause neurodegeneration [50], and 3-hydroxykynurenine (3-HK), which promotes oxidative stress and neuronal cell apoptosis [51]. Excessive production of reactive oxygen species (ROS) paired with an inadequate antioxidant response is a possible cause of inflammation, neurodegeneration, and neuronal cell death, which have all been linked to the development and progression of depression [52]. On the contrary, kynurenic acid (KYNA), which is also this pathway’s metabolite, can act neuroprotectively [53]. Studies have shown a connection between the levels of kynurenine pathway metabolites, but there is not enough evidence to support their disruptions as a possible cause of depression [43].

An alternative tryptophan metabolism route is the serotonin pathway, through which the enzyme tryptophan hydroxylase (TPH) converts tryptophan into 5-hydroxytryptophan (5-HTP), which is then converted to serotonin (5-hydroxytryptamine, 5-HT) by aromatic amino acid decarboxylase (AADC). As stated before, an increase in the activity of the kynurenine pathway, often triggered by inflammation, is associated with an imbalance between neuroprotective and neurotoxic metabolites, which can contribute to depressive symptoms. Emerging evidence suggests that even though many MDD patients may be inflammation-activated, the inflammation is often mild. If it is present, the effects of IDO activation could be masked by the up-regulation of kynurenine monooxygenase (KMO), which is also an enzyme in the kynurenine pathway and is activated by proinflammatory cytokines [54]. Both these processes could cause a shift in tryptophan metabolism away from serotonin towards kynurenine, causing serotonin levels to decline. While recent findings challenge the connection between a lower serotonin level and depression [55], serotonin is heavily associated with the gut microbiota and the gut–brain axis, which largely impacts mental health [56,57].

Omega-3 fatty acids possess anti-inflammatory properties that may help counteract the neurotoxic effects of kynurenine pathway metabolites. Evidence shows that omega-3 acids induce many effects, ranging from lowering neuroinflammation to regulating the hypothalamus–pituitary–adrenal (HPA) axis, decreasing oxidative stress and neurodegeneration, and helping maintain neuroplasticity [50]. Emerging findings suggest that *n*-3 PUFAs can modulate IDO activity and decrease the levels of toxic metabolites. Borsini et al. have found that omega-3 fatty acids can counteract the reduction in neurogenesis caused by IL-1β, a human proinflammatory cytokine, by modulating the kynurenine pathway [58]. A study by Carabelli et al. states that fish oil supplementation in animals decreases the activity of IDO and increases hippocampal levels of serotonin [59]. Research further indicates that EPA inhibits the activity of IDO through down-regulation of the protein kinase B (Akt)/mammalian target of the rapamycin (mTOR)-signaling pathway in tumor cells, which decreases kynurenine levels and increases T-cell survival. This underscores the anti-inflammatory and immunomodulatory effects of omega-3 fatty acids [60]. A study by Ilavská et al. examined the effects of omega-3 and omega-6 fatty acid supplementation on tryptophan metabolism in children and adolescents with depressive disorders. Their findings indicate that omega-3 fatty acids help modulate the kynurenine/tryptophan ratio, which may influence the kynurenine pathway activity. In contrast, omega-6 fatty acids enhance kynurenine production, which highlights their role in inflammatory processes [29]. The influence of PUFAs on serotonin levels is not fully understood.

All this suggests that omega-3 fatty acids may exert a regulatory effect on tryptophan metabolism, potentially supporting mood stability. Given that inflammation is a known contributor to mood disorders, including antenatal and postpartum depression, this interaction is particularly relevant in the perinatal period.

#### 3.2.2. Omega-3 Supplementation as a Potential Intervention for Perinatal Depression

Given the anti-inflammatory properties of omega-3 PUFAs, understanding their potential therapeutic role in perinatal depression is particularly relevant. There has been increasing interest in omega-3 supplementation as a possible strategy to reduce the prevalence of antenatal and postpartum depression.

Emerging studies have suggested that omega-3 supplementation before, during, and after pregnancy could have a positive outcome on preventing perinatal depression.

A 2018 study by Hsu et al. [61] has demonstrated that the supplementation of EPA-rich oils can reduce depressive symptoms during pregnancy and after childbirth. Long-term supplementation with DHA-rich oils could aid in reducing the risk of postpartum depression in healthy women. However, it poses no benefit for women who begin treatment while pregnant and continue after the child is born [61]. More studies have suggested that the supplementation of *n*-3 PUFAs positively impacts maternal mental health, helping diminish depressive symptoms both in pregnant women and postpartum [33]. A meta-analysis conducted by Liao et al. suggests that a higher intake of omega-3 fatty acids, with a composition of ≥60% of EPA, could have a positive outcome on alleviating depression, although it highlights the need for further studies on the exact components of the *n*-3 PUFA group [62]. A 2024 study by Li et al. has demonstrated that supplementation of omega-3 fatty acids improves depressive symptoms in adolescents [63]. As previously stated, evidence suggests that populations with an increased dietary intake of omega-3-rich fish during pregnancy are at a lower risk of postpartum depression at 6 months after pregnancy [38].

However, there are several studies stating that supplementation of omega-3 fatty acids can cause either a variable outcome on depressive disorder symptoms, or no outcome at all. A 2024 review by Serefko et al. [64] suggests that even though there is some statistical evidence of a positive impact of *n*-3 PUFAs on mood regulation, it is not identical to clinical evidence, thus advocating for a personalized approach for each patient. The patients’ diet, age, metabolic function and multiple other factors can impact the effectiveness of omega-3 fatty acid supplementation, supporting the variable outcome possibility [64]. Moreover, a therapy consisting of both antidepressant drugs and omega-3 supplements has been shown to have a greater positive influence on mood stability improvement than an antidepressant drug or a supplement alone [65].

### 3.3. Lecithin and Choline

Lecithin is a compound found naturally in various animal and plant tissues, including egg yolks, soybeans, and peanuts [66]. It is composed mainly of phospholipids, with phosphatidylcholine being the primary component. Other significant components include phosphatidylethanolamine, phosphatidylserine, and phosphatidylinositol, along with a variety of triglycerides, fatty acids, and carbohydrates [67]. 

Phosphatidylcholine, a key phospholipid within lecithin, is an essential part of cell membranes, helping to maintain their structure. It serves as a direct source of choline, an essential nutrient involved in neurotransmitter synthesis, particularly the production of acetylcholine [68]. Acetylcholine (ACh) is a neurotransmitter integral to various physiological functions, including muscle activation, attention, learning, and memory, as well as mood stability [69] (Figure 3).

During pregnancy, choline plays a vital role in ensuring the health and safety of both the mother and the developing child. It helps reduce the risk of neural tube defects and gestational complications, such as preeclampsia, gestational diabetes, and preterm birth. Moreover, it improves cognitive functions in both the mother and the child, potentially lowering the risk of developing Alzheimer’s disease later in life. Furthermore, lecithin may also help diminish the risk of developmental disorders in the child, namely autism spectrum disorder (ASD) and attention deficit hyperactivity disorder (ADHD) [70].

Choline in the human body is stored in tissues in cell membranes as phospholipids or as intracellular phosphatidylcholines and glycerophosphocholines. In the brain, it is stored mostly in the membrane-bound form, from which it can be hydrolyzed by choline acetyltransferase to provide choline for acetylcholine synthesis [71]. Free choline, phosphocholine, and glycerophosphocholine are taken up in the small intestine and transported to the liver, where they are either stored or distributed throughout the organism [72]. During pregnancy, maternal liver choline stores may deplete due to the fetal demand and the biosynthesis of acetylcholine and methyl groups in the placenta, especially during the second half of pregnancy. Despite decreased hepatic levels of choline, maternal plasma concentration increases as a consequence of enhanced transport to the fetus and various metabolic adjustments [73].

Choline metabolism demonstrates responsiveness to supplementation during pregnancy [74]. Findings suggest that prenatal choline intake promotes increased hippocampal levels of nerve growth factor (NGF), brain-derived neurotrophic factor (BDNF), vascular endothelial growth factor (VEGF), and insulin-like growth factor (IGF2) [75]. NGF promotes neuronal growth, differentiation, and survival [76]. BDNF is vital for maintaining synaptic development and plasticity [77]. Low levels of BDNF are known to correspond with higher depressive symptoms in the mother and risk for low birth weight [78]. VEGF plays an important part in embryo implantation by enhancing endometrial reactivity and promoting embryonal growth [79]. IGF2 is a vital factor in the formation and function of the placenta, as well as fetal growth during pregnancy [80].

Considering the pivotal role of choline in maintaining the health of the mother, the child, and the general population, the European Food Safety Authority established guidelines for adequate intakes (AI) of choline, namely 400 mg per day for an adult, 480 mg per day for pregnant women, and 520 per day for lactating women [71]. Although current research conducted mostly in Europe suggests that the average choline intake does not surpass 80% of AI, which is insufficient, shifting dietary patterns may be a cause for a change in this area [81]. The primary source of choline for humans is lecithin found externally, as previously mentioned, in foods of both plant (vegetable oils, soybeans) and animal (milk, egg yolks, fish) origins [82]. Lecithin is a frequently found food additive, commonly labeled E322, used as an emulsifying agent [83]. Moreover, there is a growing trend in choline supplementation, for instance, in the form of choline alfoscerate, also known as alpha-glycerophosphocholine (α-GPC, GPC), due to its presumed activity against Alzheimer’s disease and dementia [84]. This specific form of choline easily crosses the blood–brain barrier in comparison to other forms. Rodent models have shown that it enhances the release of acetylcholine in the hippocampus [85]. 

To a lesser extent, choline is produced in the body de novo in the reaction of methylation of phosphatidylethanolamine to phosphatidylcholine, catalyzed by phosphatidylethanolamine *N*-methyltransferase (PEMT). Choline is then synthesized from phosphatidylcholine using phospholipases. In the brain, de novo synthesis is very low, which is why it relies on external sources and a carrier in the blood–brain barrier [86].

#### Choline and Depression

Choline mainly acts in the human body by enhancing the synthesis, release, and availability of acetylcholine [87]. Abnormally high acetylcholine signaling has been proven to lead to depressive symptoms in human and animal models [88]. Further supporting this idea, studies have found that enhanced cholinergic signaling in the hippocampus could be linked to heightened stress response and depressive-like behaviors in rodent models [89].

The so-called acetylcholine theory of depression, which highlights the role of this neurotransmitter in the pathogenesis of depression, was first introduced by Janowsky et al. in 1974 [90]. Recent research, using modern neuroimaging techniques, such as proton magnetic resonance spectroscopy (1H-MRS), suggests that patients with major depressive disorder (MDD) may display generally disturbed levels of choline, as well as an imbalance in the choline/creatine ratio [91]. This catecholaminergic–cholinergic balance hypothesis of depression is further supported by studies indicating that treatment with acetylcholinesterase inhibitors (AChEI), such as physostigmine, could cause an increase in anxiety and depressive behaviors. Muscarinic or nicotinic acetylcholine receptor antagonists (mAChR or nAChR antagonists), such as scopolamine (mAChR antagonist), aid in alleviating these depressive symptoms. Other drugs that showed a tendency towards diminishing the negative symptoms were ketamine, even among drug-resistant patients, and fluoxetine, a selective serotonin reuptake inhibitor (SSRI) [89,92]. Ketamine has been proven to decrease ACh levels in various regions of the brain in rodent models [93].

A recent study, which analyzed circulating diet-sensitive metabolites presenting a significant correlation with depression, observed a higher level of lecithin in individuals with depression [94]. A study conducted by van Lee et al. on 949 women at 26 to 28 weeks of gestation suggests that higher choline levels during pregnancy are correlated with depression and anxiety symptoms, which were measured in Edinburgh Postnatal Depression Scale (EPDS) and State-Trait Anxiety Inventory (STAI), respectively. On the contrary, at 3 months postpartum, no significant association between choline levels and the mental state of the patients was detected [95]. 

As stated before, the gut–brain axis plays an important role in mental health, and choline is present in this process as well. Choline is metabolized by the gut microbiota into trimethylamine (TMA) by specific enzymes, such as choline TMA-lyase. The majority of TMA formed in the intestinal tract and absorbed into the portal circulation is oxidized in the liver by flavin-containing monooxygenase (FMO), primarily FMO3, in order to produce trimethylamine *N*-oxide (TMAO) [96]. TMAO can cross the blood–brain barrier and influence the brain’s function, which correlates with the finding that serum concentration of TMAO is linked to depression severity [97]. A study comparing serum levels of TMAO within individuals with depression identified a significant positive correlation to the intensity of depressive symptoms among men without carbohydrate malabsorption [98]. TMAO has also been identified as a potential biomarker for stress and depressive symptoms after both stroke and myocardial infarction, which highlights its importance in the pathophysiology of depression [99,100].

There is growing evidence supporting the role of choline and phosphatidylcholine in MDD, as well as perinatal depression (Table 2). However, specific evidence remains inconclusive and further research is essential to investigate this phenomenon, especially within the population of women in the perinatal period. These findings could contribute to identifying potential biomarkers and result in dietary recommendations in the case of excessive consumption.

The studies included in this narrative review demonstrated heterogeneous methodological quality. Most recent clinical and observational studies (2020–2025) provided moderate to strong evidence on the role of omega-3 PUFAs in perinatal depression. Potential sources of bias primarily relate to small sample sizes, heterogeneous diagnostic criteria for depression, dietary self-reporting, and confounding variables such as socioeconomic status, comorbidities, or concurrent supplementation. While omega-3 fatty acids are supported by moderate to strong evidence for preventive and therapeutic potential in perinatal depression, the data on lecithin and choline remain preliminary. Future well-controlled, large-scale longitudinal studies are needed to minimize confounding effects and confirm causal relationships (Table 3).

## 4. Conclusions

Perinatal depression is a major public health issue with complex causes. The evidence reviewed in this study suggests that changes in lipid metabolism may be an important factor in the pathophysiology of antenatal and postpartum depression. Research shows that fatty acids, especially omega-3 PUFAs, may play a key role in mood regulation and managing inflammation during pregnancy and postpartum. Lower levels of these fatty acids appear to increase the risk of mood disorders during the perinatal period. Additionally, choline and its metabolites, such as TMAO, are involved in neurotransmitter regulation and inflammatory processes, linking them to depressive symptoms. 

Although several studies indicate potential benefits of omega-3 PUFA or choline supplementation, the overall findings remain inconsistent due to differences in study design, dosage, timing, and population characteristics (Appendix A). The positive effects of supplementation appear to depend on the specific fatty acid composition and timing of administration. EPA-rich formulations tend to reduce depressive symptoms during pregnancy and postpartum, while DHA-rich supplements may help prevent postpartum depression when used as long-term supplementation in otherwise healthy women. However, not all trials confirm clinical improvement in depressive symptoms, as there are many variables to be considered, such as the woman’s diet, age or metabolic function. Moreover, excessive or unbalanced supplementation might alter metabolic or inflammatory pathways in unpredictable ways. Therefore, a personalized approach to supplementation should be considered over generalized recommendations.

From a clinical perspective, these findings underscore the importance of monitoring lipid and choline metabolism during pregnancy as potential modulators of maternal health. Early identification of women at risk and nutritional optimization may contribute to prevention strategies.

Future research should focus on large-scale, well-controlled longitudinal studies and randomized clinical trials integrating biochemical, hormonal, and genetic markers. Such studies are essential to clarify causal relationships, as well as establish optimal supplementation protocols and define which subgroups of women are most likely to benefit.

## Figures and Tables

**Figure 1 nutrients-17-03590-f001:**
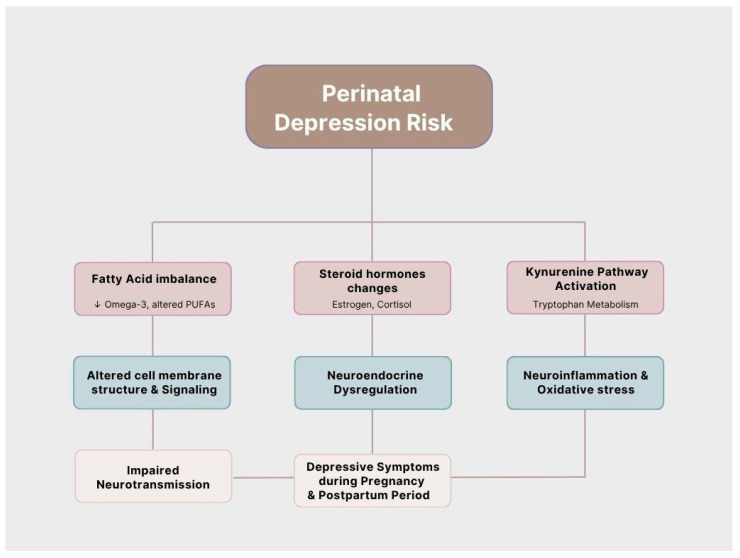
Overview of key lipid-metabolism factors influencing the risk of perinatal depression.

**Figure 2 nutrients-17-03590-f002:**
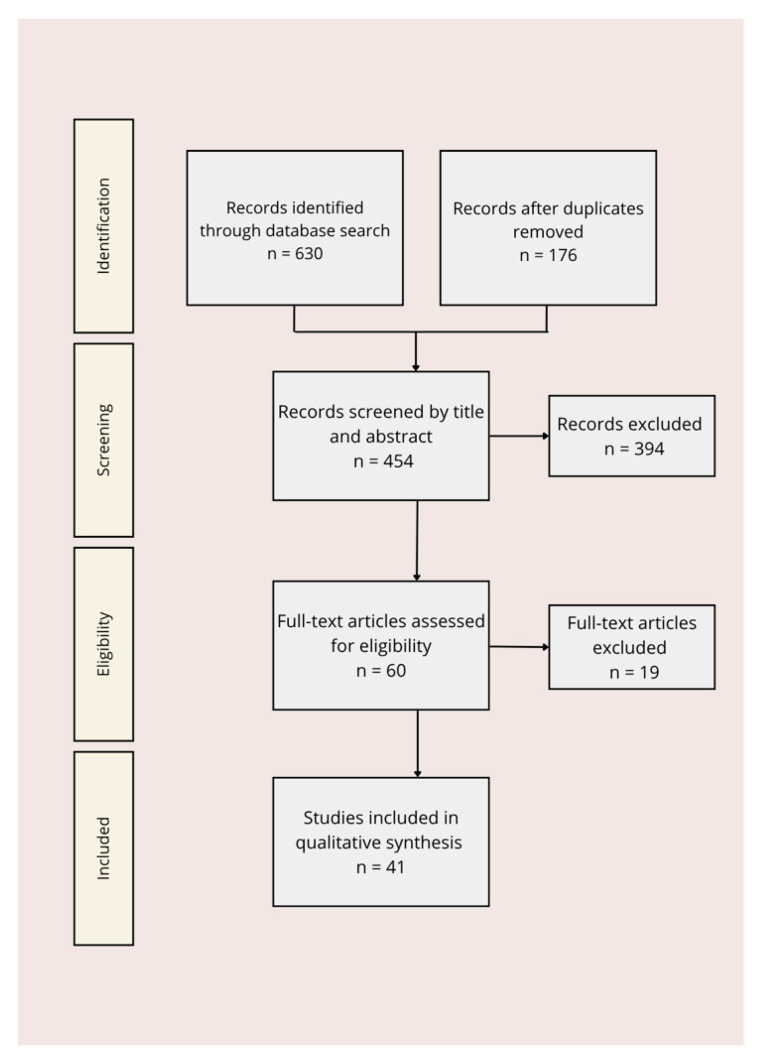
PRISMA flow diagram of study selection.

**Figure 3 nutrients-17-03590-f003:**
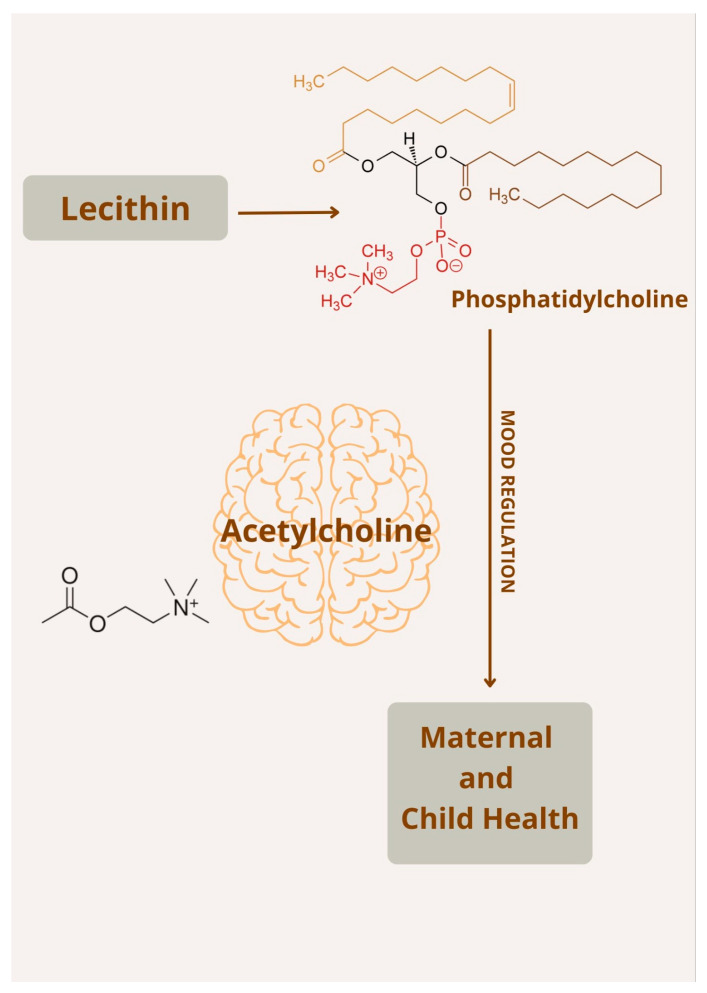
Overview of lecithin metabolism and its significance for maternal and fetal brain function.

**Table 1 nutrients-17-03590-t001:** Overview of Major Fatty Acids and Their Role in Perinatal Depression. PUFA—polyunsaturated fatty acid; BBB—blood–brain barrier.

Fatty Acid	Type	Symbol	Physiological Role	Brain Presence/Function	Association with Perinatal Depression
α-Linolenic Acid (ALA)	Omega-3 PUFA	18:3*n*-3	Precursor of EPA and DHA	Crosses BBB but quickly oxidized; low proportion in brain lipids	Deficiency limits DHA synthesis, indirectly increasing depression risk
Eicosapentaenoic Acid (EPA)	Omega-3 PUFA	20:5*n*-3	Anti-inflammatory, supports neurotransmission	Crosses BBB; limited accumulation	Higher EPA/DHA ratio (≥1.5) associated with improved mood; deficiency linked to depressive symptoms
Docosahexaenoic Acid (DHA)	Omega-3 PUFA	22:6*n*-3	Essential for neuronal membranes, synaptic function, neurogenesis	Major *n*-3 PUFA in grey matter; accumulates in fetal brain	Low maternal DHA linked to increased risk of antenatal and postpartum depression
Docosapentaenoic Acid (DPA)	Omega-3 PUFA	22:5*n*-3	Intermediate between EPA and DHA	Crosses BBB but rapidly oxidized	Limited evidence; may support DHA-related neuroprotection
Linoleic Acid (LA)	Omega-6 PUFA	18:2*n*-6	Precursor of arachidonic acid (AA)	Supports cell membrane integrity	Imbalance (excess omega-6) increases inflammation and depression risk
Arachidonic Acid (AA)	Omega-6 PUFA	20:4*n*-6	Inflammatory mediator, signal transduction	Major *n*-6 PUFA in brain grey matter	High AA/DHA ratio associated with impaired synaptic function and depressive symptoms

**Table 2 nutrients-17-03590-t002:** Characteristcs and key findings of original studies and reviews examining cholinergic mechanisms, TMAO, and lecithine in relation to mood and depression.

First Author (Year)	Ref.	Country/Setting	Population	Study Design	Key Findings
Mineur (2013)	[89]	USA	Mice	Preclinical in vivo	Higher acetylocholine levels in the hippocampus are linked to more anxiety- and depression-like behaviour; this effect is reversible with fluoxetine
Janowsky (1974)	[90]	USA	Adults	Clinical challenge experiment	Central cholinergic stymulation provoked depressive symptoms
Riley (2018)	[91]	USA	-	Narrative review	Most neuroimaging studies show elevated choline levels in major depressive disorder (MDD)
Dulawa (2018)	[92]	USA	-	Narrative review	Increased cholinergic activity linked to low mood; AchE inhibitors often worsen depressive symptoms
Hong (1987)	[93]	South Korea	Rats	Preclinical in vivo	Ketamine has the capacity to alter acetocholine levels
van der Spek (2023)	[94]	Multi-Country European Cohorts	Adults (*n* = 13,596)	Multi-cohort observational study	Altered diet-related metabolites in depression with elevated levels of lecithine and reduced SCFAs concentration
van Lee (2017)	[95]	Singapore (Gusto Cohort)	Pregnant women (*n* = 949)	Prospective cohort	Higher maternal choline linked to more antenatal depressive/anxiety symptoms
Mudimela (2022)	[96]	International	-	Narrative review	TMAO promotes neuroinflammation via oxidative stress and micoglial activation which can lead to psychiatric disoreders
Liu (2023)	[97]	International	-	Narrative review	Altered TMAO, SCFAs, kynurenine pathway activation linked to gut dysbosis with neuroinflammation and depression
Meinitzer (2020)	[98]	Austria	Adults (*n* = 251)	Cross-sectional observational	Increased TMAO linked to depressive symptoms; zonulin associations were sex-specific
Hu (2024)	[99]	China	Rodent model; cellular assays	Preclinical in vivo	TMAO expression aggreviated post-stroke depression, increased brain blood barrier (BBB) permeability and decreased neutrophic signalling
Von Lewinski (2021)	[100]	Austria	Myocardial infarction (MI) patients (*n* = 52)	Observational cohort	Higher TMAO after myocardial infarction linked to severe perceived stress; potential stress biomarker

**Table 3 nutrients-17-03590-t003:** Comparative strength of the included evidence.

Aspect	Type of Evidence	Strengh of Evidence	Main Limitations
Omega-3 PUFAS and Perinatal Depression	Randomized controlled trials, cohort studies	Moderate to strong	Heterogeneous supplementation doses and formulations Variable control for diet and inflammation markers
Lecithin and Choline in Perinatal Mood Regulation	Observational and animal studies	Weak to moderate	Small sample sizes Inconsistent measures of choline status Lack of standardized outcomes
Inflammatory and Kynurenine Pathways	Mechanistic and clinical studies	Moderate	Limited human data Potential confounding by nutrition and stress variables
Supplementation Efficacy (EPA vs. DHA Ratio)	Meta-analyses, clinical trials	Moderate	Short study duration Publication bias Self-reported compliance
Overall Lipidome–Mood Associations	Integrative reviews, metabolomic studies	Moderate	High heterogeneity Cross-sectional designs Causality not established

## Data Availability

No new data were created or analyzed in this study, data Sharing is not applicable to this article.

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
