# Peer review of "Selected Lipidome Components and Their Association with Perinatal Depression"

_nutrients, 2025, doi:10.3390/nu17223590_

Round 1

Reviewer 1 Report

Comments and Suggestions for Authors

This is a well-written and well-structured narrative review exploring the relationship between lipid metabolism (particularly omega-3 fatty acids, lecithin, and choline) and perinatal depression.  The manuscript demonstrates strong writing quality, a coherent flow, and an up-to-date literature base extending through 2025.

Some points in the article that need to be addressed:

  • Need for Critical Synthesis:
    The review is primarily descriptive. It would benefit from a stronger critical comparison across studies — e.g., highlighting methodological strengths/weaknesses, sample size limitations, or inconsistent findings.
    Suggestion: Add a short subsection in Section 3 (“Results and Discussion”) summarizing the comparative strength of evidence or potential bias sources (perhaps in tabular form).

  • Transparency in Methodology:
    Although a “structured search” is mentioned, the search string, inclusion process, and study flow are not reported.
    Suggestion: Include a PRISMA-style flow diagram or a brief description of the screening and selection process in the Appendix.

  • Choline/Lecithin Evidence Base:
    The discussion acknowledges limited data, but the section would be strengthened by summarizing the few available studies in a concise table (sample size, main findings, outcomes).

  • Language and Consistency:

    • Minor typographical issues (e.g., “Results:.” → double punctuation).

    • Unify British/American spelling and ensure abbreviation consistency (PUFA, DHA, EPA, etc.).

    • Review minor formatting issues (e.g., missing parenthesis in “Type of the Paper (Article”).

  • Clinical Relevance:
    The conclusion section could be enhanced by briefly discussing clinical or dietary implications (e.g., potential for omega-3 screening or supplementation recommendations in perinatal care).

Author Response

Dear Reviewer,  

Thank you for your valuable remarks and your effort to review the manuscript.  

Our point-by-point response is below:  

Need for Critical Synthesis:
The review is primarily descriptive. It would benefit from a stronger critical comparison across studies — e.g., highlighting methodological strengths/weaknesses, sample size limitations, or inconsistent findings.
➤ Suggestion: Add a short subsection in Section 3 (“Results and Discussion”) summarizing the comparative strength of evidence or potential bias sources (perhaps in tabular form).
We have added a table comparing the strength of evidence and the potential risk of bias

Transparency in Methodology:
Although a “structured search” is mentioned, the search string, inclusion process, and study flow are not reported.
➤ Suggestion: Include a PRISMA-style flow diagram or a brief description of the screening and selection process in the Appendix.

We have added a PRISMA-style flow diagram in order to ensure transparent reporting of study identification, screening, eligibility, and inclusion.

Choline/Lecithin Evidence Base:
The discussion acknowledges limited data, but the section would be strengthened by summarizing the few available studies in a concise table (sample size, main findings, outcomes).

The evidence on this topic remains limited, although it is a potentially important direction for future research. To strengthen the summary of the existing findings, we have added a table presenting the key characteristics of the included studies.

Language and Consistency:
o Minor typographical issues (e.g., “Results:.” → double punctuation). o Unify British/American spelling and ensure abbreviation consistency (PUFA, DHA, EPA, etc.).
o Review minor formatting issues (e.g., missing parenthesis in “Type of the Paper (Article”).

We have thoroughly revised the manuscript to correct typographical errors and formatting issues. While we have made every effort to ensure accuracy, we remain open to any additional editorial suggestions.

  • Clinical Relevance:
    The conclusion section could be enhanced by briefly discussing clinical or dietary implications (e.g., potential for omega-3 screening or supplementation recommendations in perinatal care).  

In accordance with the reviewer’s suggestion, we expanded the “conclusions” section to highlight the clinical implications and the potential directions for future research. 

We sincerely appreciate the time and effort you have taken to review our manuscript.  

  

Best regards,  

Authors.  

Reviewer 2 Report

Comments and Suggestions for Authors

Overall, the paper is well-written, both linguistically and in terms of conceptual organization.
Below, I outline some points that I consider particularly relevant regarding the structure and content of the paper, with the aim of further enhancing its clarity, impact, and scientific value.

Materials and Methods: Although this is a narrative review, since the authors specified the databases used and the keywords, it would be helpful to report the number of papers identified, included, or excluded, as well as the number of meta-analyses, reviews, and research papers considered. This would provide readers with a clearer understanding of the context from which the paper’s content emerges.

The title refers to the lipidome; however, some aspects of the lipidome appear to be only superficially addressed (for instance, ceramides and phospholipids other than lecithin). Did the authors not identify articles on these topics, or could the search parameters have influenced the results? Alternatively, was the review specifically focused on Omega-3 and lecithin/choline?

Overall, the concluding section could be strengthened by including a more critical discussion of the findings, highlighting both the positive and negative aspects of certain supplementation strategies. It would also be valuable to emphasize the clinical implications of the results and identify areas where further research is warranted.

Before the references, there is an Appendix A. However, I could not find a clear reference in the text specifying which data are included in Appendix A or what the cited articles refer to. It seems that the review covers a larger number of papers—could the authors clarify this?

Minor points:

  • Line 84: Figure 1. The caption reads “key factors”; it would be clearer to specify that these are key factors within the scope of this paper (i.e., lipid metabolism). Many other factors, such as socio-environmental influences, can also contribute to the onset of perinatal depression.

  • Lines 141-143: A reference is suggested here.

Author Response

Dear Reviewer,

Thank you for your valuable remarks and your effort to review the manuscript.

Our point-by-point response is below:

Overall, the paper is well-written, both linguistically and in terms of conceptual organization. Below, I outline some points that I consider particularly relevant regarding the structure and content of the paper, with the aim of further enhancing its clarity, impact, and scientific value.

Materials and Methods: Although this is a narrative review, since the authors specified the databases used and the keywords, it would be helpful to report the number of papers identified, included, or excluded, as well as the number of meta- analyses, reviews, and research papers considered. This would provide readers with a clearer understanding of the context from which the paper’s content emerges.

We have now provided the total number of studies that we thoroughly read and analyzed, with a clear distinction between those used to build the theoretical background and those contributing to the narrative synthesis. In addition, we specified the types of studies included.

The title refers to the lipidome; however, some aspects of the lipidome appear to be only superficially addressed (for instance, ceramides and phospholipids other than lecithin). Did the authors not identify articles on these topics, or could the search parameters have influenced the results? Alternatively, was the review specifically focused on Omega-3 and lecithin/choline?

The narrative review was primarily focused on lecithin and Omega-3, therefore, we have modified the title in order to more accurately reflect its scope and to avoid any potential misinterpretation. The revised title is: “Selected lipidome components and their association with perinatal depression”.

Overall, the concluding section could be strengthened by including a more critical discussionof the findings, highlighting both the positive and negative aspects of certain supplementationstrategies. It would also be valuable to emphasize the clinical implications of the results and identify areas where further research is warranted.

As previously explained.

Before the references, there is an Appendix A. However, I could not find a clear reference in the text specifying which data are included in Appendix A or what the cited articles refer to. It seems that the review covers a larger number of papers— could the authors clarify this?

Appendix A contains a comprehensive summary table of major cited studies included in the narrative review (e.g., Rees et al., 2009; Hoge et al., 2019; Mocking et al., 2020; Ilavska et al., 2024). The table provides key methodological and contextual details for each study. The purpose of including Appendix A is to enhance transparency and allow readers to quickly identify the source, design, and main outcomes of important studies referenced in the text.

Minor points:

Line 84: Figure 1. The caption reads “key factors”; it would be clearer to specify that these are key factors within the scope of this paper (i.e., lipid metabolism). Many other factors, such as socio-environmental influences, can also contribute to the onset of perinatal depression.

We have corrected the above-mentioned sentence to: “Overview of key lipid-metabolism factors influencing the risk of perinatal depression”.

Lines 141-143: A reference is suggested here. We decided to remove this fragment, as we were unable to locate the original source and we considered it more appropriate not to include information that cannot be supported by a verified reference.

We sincerely appreciate the time and effort you have taken to review our manuscript.

Best regards,

Authors.

Round 2

Reviewer 2 Report

Comments and Suggestions for Authors

I suggest the authors to explain what Appendix A refers to.

Beyond this, the paper is suitable for publication

Congratulations to the authors